# Transporter-Targeted Nano-Sized Vehicles for Enhanced and Site-Specific Drug Delivery

**DOI:** 10.3390/cancers12102837

**Published:** 2020-10-01

**Authors:** Longfa Kou, Qing Yao, Hailin Zhang, Maoping Chu, Yangzom D. Bhutia, Ruijie Chen, Vadivel Ganapathy

**Affiliations:** 1Department of Pharmacy, The Second Affiliated Hospital and Yuying Children’s Hospital of Wenzhou Medical University, Zhejiang 325027, China; koulongfa@wmu.edu.cn; 2Wenzhou Municipal Key Laboratory of Pediatric Pharmacy, Zhejiang 325027, China; yaoqing@wmu.edu.cn (Q.Y.); 194048@wzhealth.com (H.Z.); chmping@wzhealth.com (M.C.); 3Department of Pharmaceutical Sciences, Wenzhou Medical University, Zhejiang 325035, China; 4Department of Children’s Respiration Disease, The Second Affiliated Hospital and Yuying Children’s Hospital of Wenzhou Medical University, Zhejiang 325027, China; 5Pediatric Research Institute, The Second Affiliated Hospital and Yuying Children’s Hospital of Wenzhou Medical University, Zhejiang 325027, China; 6Department of Cell Biology and Biochemistry, Texas Tech University Health Sciences Center, Lubbock, TX 79430, USA; yangzom.d.bhutia@ttuhsc.edu

**Keywords:** plasma membrane transporters, nano-drug delivery systems, targeted drug delivery, oral absorption, site-specific drug delivery

## Abstract

**Simple Summary:**

Nano-drug delivery systems serve as Trojan horses to carry therapeutic drugs as cargos and deliver them to target cells. The specificity of these delivery vehicles to a particular cell type can be improved if the surface of the vehicles is chemically modified in such a manner that they are recognized and attracted to the target cell. This can be accomplished if the target cell selectively expresses a receptor or transporter and the surface of the drug-delivery vehicles is conjugated with the receptor agonist or the transporter substrate. In this review, we detail published literature on the successful exploitation of plasma membrane transporters for this purpose. In particular, this review emphasizes the delivery of chemotherapeutic drugs to cancer cells by targeting the nano-delivery systems specifically to certain transporters that are selectively upregulated in cancer cells.

**Abstract:**

Nano-devices are recognized as increasingly attractive to deliver therapeutics to target cells. The specificity of this approach can be improved by modifying the surface of the delivery vehicles such that they are recognized by the target cells. In the past, cell-surface receptors were exploited for this purpose, but plasma membrane transporters also hold similar potential. Selective transporters are often highly expressed in biological barriers (e.g., intestinal barrier, blood–brain barrier, and blood–retinal barrier) in a site-specific manner, and play a key role in the vectorial transfer of nutrients. Similarly, selective transporters are also overexpressed in the plasma membrane of specific cell types under pathological states to meet the biological needs demanded by such conditions. Nano-drug delivery systems could be strategically modified to make them recognizable by these transporters to enhance the transfer of drugs across the biological barriers or to selectively expose specific cell types to therapeutic drugs. Here, we provide a comprehensive review and detailed evaluation of the recent advances in the field of transporter-targeted nano-drug delivery systems. We specifically focus on areas related to intestinal absorption, transfer across blood–brain barrier, tumor-cell selective targeting, ocular drug delivery, identification of the transporters appropriate for this purpose, and details of the rationale for the approach.

## 1. Introduction

Nano-drug delivery systems (NDDS) have attracted a lot of attention in recent years due to their unique characteristics, including increased drug solubility and stability, sustained drug release, prolonged life in circulation, reduced toxicity, and enhanced therapeutic efficacy [1,2,3]. The marked progress in the development of biomaterials and novel nanotechnology strategies has added further fuel to the advancement in the field [4,5,6,7,8]. The results have been encouraging, with several drugs in nano-formulations already in use in the clinics (e.g., Lipusu, Doxil, Abraxane, and Genexol-PM). The versatility of NDDS broadens the scope of their applications. They can be used to optimize the delivery of drugs belonging to multiple chemical groups: small-molecules, peptides, proteins, RNA, and DNA. When incorporated into NDDS, the drugs generally display pharmacokinetic and pharmacodynamic characteristics that are very different from those of their original forms, a feature contributing to the increased interest and utilization of NDDS in the clinics [9,10]. Ideally, therapeutic agents are expected to reach their sites of action across multiple biological barriers, including the intestinal tract, blood–brain barrier (BBB), and blood–tumor barrier and also to target specific cell types or tissues, thus minimizing unwanted off-target effects. Even though NDDS have shown promise in some of these areas, it is far from satisfactory and the field is still in its infancy and constantly evolving with better results. One of the strategies contributing to these improved outcomes involves chemical modification of the surface of NDDS with specific ligands, which can be recognized by cell-surface components that are expressed preferentially on target cells. Cell-surface receptors have been used for targeted drug delivery in preclinical studies for decades, but unfortunately, there is still no product in this category that has successfully moved into the clinics. One biggest issue is the unpredictable targeting efficacy, which is oftentimes not uniform among different individuals due to the high variability and heterogeneity in the expression of these receptors [11,12,13]. Therefore, there is an urgent need to identify new strategies to improve the clinical utility of NDDS.

Plasma membrane transporters are a special group of integral transmembrane proteins, which play an obligatory role in providing mammalian cells specific nutrients such as glucose, amino acids, vitamins, trace elements (e.g., iron and zinc), and purine/pyrimidine bases and nucleosides. The biology and metabolic needs unique to each cell type dictate which specific transporters need to be expressed on its cell surface. Furthermore, it is not just the types of transporters that differ from cell type to cell type; it is also the density of these transporters on the cell surface because the magnitude of the needs for a given nutrient is not the same for all cell types. More than 400 transporters have been discovered in human cells, and these transporters could be classified into two families: ATP-binding cassette (ABC) transporters and solute carriers (SLC) [14]. The recognition and transport capability of these transporters are not restricted to their physiologic substrates. A wide variety of therapeutics are recognized and transported by these transporters, most often based on the extent of structural similarities between the drugs and the physiologic substrates. As such, the plasma membrane transporters function as key determinants in the absorption, distribution, and excretion of drugs [14,15]. ABC transporters in mammalian cells are exclusively exporters for their endogenous and exogenous substrates from the cells coupled to ATP hydrolysis on the cytoplasmic side as the driving force [16,17]. SLC transporters, on the other hand, are more involved in the absorption and distribution of nutrients and drugs because they mediate either the influx or efflux of their substrates depending on the involvement and direction of driving forces, which are ionic gradients and membrane potential. Because of the marked influence of the ABC transporters and SLC transporters in the handling and therapeutic efficacy of almost all drugs, it is a prerequisite at present that all drugs in the development process should be investigated for their interaction with a selective group of transporters prior to approval by the Food and Drug Administration (FDA). As most often it is the structural similarity to the natural substrate that determines the interaction of a drug with any given transporter, chemically modifying the drugs could facilitate such interaction, thereby increasing the absorption and distribution of the drugs to maximize their therapeutic efficacy. This is the basis of the prodrug approach. Examples of the success of this approach include valacyclovir [18] and valganciclovir [19]. The parent drugs, acyclovir and ganciclovir, exhibit poor oral bioavailability, but when chemically modified with the amino acid valine, the resultant derivatives are recognized by the peptide transporter PEPT1 in the intestinal tract and absorbed efficiently.

In recent years, the plasma membrane transporters are emerging as novel targets in the development of NDDS for the efficient delivery of their drug cargos [20,21]. Several transporters are expressed in the intestinal tract for the absorption of a wide variety of nutrients of diverse chemical nature such as monosaccharides, amino acids, small peptides, vitamins, bacterial fermentation products, and trace metals. Some of them, including the peptide transporter PEPT1 [22] and the vitamin C transporter SVCT1 [23], have been reported as potential nano-drug delivery targets to facilitate oral absorption. Similarly, selective transporters are highly expressed at blood–brain barrier to provide necessary nutrients to brain cells to support survival and growth; such transporters have been found to be useful as targets for brain drug delivery. Examples include the facilitative glucose transporter GLUT1 [24] and the choline transporter ChT1 [25]. Tumor cells have an increased demand for nutrients to support the malignant proliferation, and therefore upregulate selective nutrient transporters to meet this need. Thus, the abnormally over-expressed transporters in the tumor tissue also provide ideal targets for selective drug delivery in cancer treatment [26,27]. Therein, the expression characteristics of the transporters in the biological barrier sites and other cell types under pathological conditions have transformed these transporters as potential targets for enhanced and site-specific drug delivery.

In recent years, an increasing number of transporter-targeted NDDS have been designed and investigated to improve oral absorption, increase brain distribution, and enhance the accumulation at disease sites, of several therapeutic drugs. These studies verify the potential of transporter-targeted NDDS in various therapeutic applications. In this review, we summarize recent advances in the field of transporter-targeted NDDS for multiple purposes, including improved oral absorption, tumor-targeted delivery, transfer across BBB, and topical ocular delivery (Figure 1). We also offer some perspectives and comments for the future direction of transporter-targeted NDDS.

## 2. Transporter-Targeted NDDS for Improved Oral Absorption

Oral administration is the most convenient route for most patients, and this mode of drug intake increases the chances of compliance. However, many therapeutic agents have poor oral absorption and bioavailability due to water insolubility, weak permeability, inferior stability, and the unsuitable luminal environment in the gastrointestinal tract. It is therefore necessary to develop effective strategies to increase the oral bioavailability of these drugs. The transporters expressed in the gastrointestinal tract play an important role in the absorption of dietary nutrients. Transporter-assisted oral drug delivery in the form of prodrugs has a successful track record [18,19]. Recently, there have been an increasing number of published reports on the use of transporters in the intestinal tract for the delivery of drugs in the form of NDDS. In this section, we will discuss the benefits and current challenges in this area related to the design and use of NDDS for the promotion of oral absorption of drugs.

### 2.1. Increasing the Oral Absorption of Small Molecules

The therapeutic drugs that are included in the biopharmaceutical classification system (BCS) IV usually display low water solubility and poor permeability, resulting in decreased oral bioavailability. Unfortunately, many anticancer therapeutics belong to this category. Additionally, extreme acidity in the gastric lumen, exposure to digestive enzymes and potentially consequent degradation in the intestinal lumen, and active efflux of absorbed drugs back into the lumen via export transporters such as P-glycoprotein and ABCG2 (breast cancer resistance protein) in enterocytes contribute to the decreased oral bioavailability of such drugs. Encapsulating these drugs into nanocarriers could increase their solubility and stability, but the permeability across the intestinal barrier is still a hurdle. Transporter-targeted NDDS could address this issue by targeting the plasma membrane transporters that are expressed in the absorptive cells (enterocytes) of the intestinal tract and facilitate their oral bioavailability and entry into systemic circulation across the intestinal barrier.

PepT1/SLC15A1 is a transporter with high capacity and low affinity for dipeptides and tripeptides derived from the breakdown of dietary proteins; this transporter is highly expressed along the entire small intestine [28]. PepT1-targeted nanoparticles have been developed and evaluated. Du et al. [22] conjugated the dipeptides L-valyl-L-valine and L-valyl-L-phenylalanine to polyoxyethylene stearate to modify poly (lactic-co-glycolic acid) (PLGA) NPs for enhanced oral absorption of docetaxel. Their studies showed that cellular uptake of both dipeptide-modified NPs was much higher than that of bare NPs in PepT1-expressing HeLa cells and Caco-2 cells. The L-valyl-L-valine-modified (NSPV1000) NPs showed the highest intestinal permeability; the bioavailability of NSPV1000 NPs was 4.2- and 2.0-fold higher than that of the parent drug and unmodified NPs, respectively. In addition, they found that the internalization of the dipeptide-modified NPs in the enterocytes was an H^+^-dependent process, as would be expected of PepT1, whose transport function is coupled to a transmembrane electrochemical gradient for H^+^ [28,29]. Using the same strategy, Gourdon et al. [30] conjugated valine, glycylsarcosine, valyl-glycine, and tyrosyl-valine to the synthetic polymer PLA-PEG, and prepared ligand-conjugated nanoparticles by classic nanoprecipitation method. They performed the competition assay between functionalized nanoparticles and [^3^H]glycylsarcosine, a substrate for PepT1, for uptake in Caco-2 cells. These studies showed that the dipeptide-conjugated nanoparticles could inhibit the uptake of [^3^H]-glycylsarcosine, and valine-conjugated nanoparticles displayed the strongest inhibitory effect on the uptake of [^3^H]glycylsarcosine. The in vivo pharmacokinetic study showed that valine-conjugated nanoparticles did not affect AUC or C_max_ statistically, but increased t_1/2_ and MRT compared to free drug cargo (acyclovir). In a follow-up study, they further investigated the effect of the manufacturing process on the PepT1-targeted delivery efficacy [31]. The valine-conjugated NPs were prepared by three different methods, including nanoprecipitation, simple emulsion, and double emulsion, with median particle size < 200 nm. The competition study and uptake assay demonstrated that the preparation methods had a significant impact on the transporter-targeted efficacy, and the method of double emulsion showed the strongest PetT1 targeting ability.

The intestinal tract efficiently absorbs dietary nutrients from the lumen into the portal blood. Glucose is absorbed via the sodium-coupled glucose transporter SGLT1 expressed on the apical side of the enterocytes and then translocated across the basolateral membrane via the facilitative glucose transporter GLUT2 [32]. Interestingly, GLUT2 located at the basolateral side of enterocytes has been explored as a potential target for oral biomacromolecule delivery while SGLT1-targeted NDDS has not yet been reported. It has been found that GLUT2 could be rapidly recruited from the cytosol to the apical membrane when the luminal glucose concentrations are elevated, even though GLUT2 is normally present in the basolateral side [33]. Mace et al. [34] showed that low concentrations of artificial sweeteners can effectively induce the recruitment and density of GLUT2 in the apical membrane. Wu et al. [35] designed a co-administration strategy for calorie-free sweetener acesulfame potassium and fructose-modified nanoparticles (Fru-PEG NPs) to facilitate GLUT2-mediated transcytosis for enhanced oral bioavailability (Figure 2). The combination strategy displayed an 8.8-fold increase in trans-epithelial transport in the intestine in vitro and a 3.9-fold increase in oral absorption in mice in vivo compared to regular PEGylated nanoparticles. Acesulfame potassium (AceK) could increase GLUT2 density in the apical membrane of intestinal epithelial cells, thus amplifying the transcytosis of nanoparticles from the lumen to the serosal side. Significantly increased internalization and basolateral exocytosis of Fru-PEG NPs via GLUT2 were observed. This was the first report on the facilitation of the unidirectional trans-epithelial transport of orally administered NDDS by manipulating and amplifying a nutrient-absorption pathway.

L-Carnitine is a highly polar zwitterionic compound, which is obligatory for the entry of long-chain fatty acids into the mitochondrial matrix across the inner mitochondrial membrane for subsequent β-oxidation. L-Carnitine is synthesized endogenously to a significant extent, but diet is still the main source of L-carnitine. The dietary L-carnitine is transported across the intestinal epithelium via the organic cation/carnitine transporter 2 (OCTN2/SLC22A5). Recently, OCTN2 has been used as a target for enhanced oral drug delivery, including prodrugs [36] and NDDS [37,38]. Our group first constructed L-carnitine-conjugated nanoparticles to improve the oral absorption of paclitaxel and noted a 3-fold increase in bioavailability compared to that of bare nanoparticles [39]. The cellular uptake and oral absorption of L-carnitine-conjugated nanoparticles increased as the ligand density increased from 5% to 10%, indicating that a multivalent crosslink between nanoparticles and transporters contributed to the enhanced oral absorption. However, any further increase in ligand density decreased the uptake efficacy, indicating that there is an optimal density for maximal effect. This interesting result underscores that increasing the density of the conjugated ligand on the surface of the nanoparticles might not always increase oral absorption. In the following study, we explored the possibility of increasing the flexibility of conjugated L-carnitine by different lengths of PEGs to enhance oral absorption [40]. These studies showed that the insertion of the PEG linker improved the absorption in vitro and in situ, but unexpectedly interfered with absorption in vivo. The low bioavailability of PEG-linked OCTN2-targeted NPs probably occurred either because the extended targeting length of NPs decreased the mobility and permeation or because the inserted PEG linker affected some other biological function in the absorptive cells, thus interfering with the OCTN2-mediated endocytosis. This phenomenon was also observed by Xie et al. [41]. L-Carnitine-conjugated micelles have also been developed to enhance the oral absorption of multiple water-insoluble drugs [42,43]. L-Carnitine is also a substrate for OCTN1, which is expressed in enterocytes along with OCTN2, but the affinity of OCTN1 towards L-carnitine is much lower than that of OCTN2. Therefore, it is unlikely that OCTN1 was involved in the oral absorption of L-carnitine-conjugated nanoparticles under the experimental conditions employed in our studies, but the validity of this assumption remains to be tested.

ASBT/SLC10A2 is expressed on the apical membrane of enterocytes in the ileum, which participates in the absorption of bile acids, an essential component in the enterohepatic circulation of bile acids [44]. Bile acids take part in the digestion and absorption of dietary fat and fat-soluble vitamins; these are synthesized in the liver from cholesterol and secreted into bile to aid fat digestion in the intestinal tract [45]. ASBT is an efficient and high-capacity transporter for bile acids, evidenced by the efficacy of the transporter in successfully accomplishing the intestinal absorption of ~25 g of bile acids per day with only <5% loss in the feces [44]. The expression profile and functional features of ASBT suggest that it is an ideal target for oral drug delivery. Khatun et al. [46] synthesized taurocholic acid-linked heparin-docetaxel conjugates for oral delivery of the anticancer drug. The ternary biomolecular conjugates formed self-assembly nanoparticles with docetaxel in the core and taurocholic acid on the surface of nanoparticles. These nanoparticles displayed significantly increased oral absorption and enhanced anticancer effect in mice bearing tumors arising from MDA-MB231 and KB cells; this increased efficacy was attributed to the specific interaction between nanoparticles and ASBT in the small intestine. Yin et al. [47] modified polymer-lipid hybrid nanoparticles with cholate (cPLNs) to enhance the oral absorption of quercetin for antileukemic therapy by targeting ASBT. cPLNs increased the cellular uptake and internalization capability of quercetin. In in vivo studies, quercetin-loaded cPLNs exhibited enhanced oral bioavailability and antileukemic effect compared to free quercetin and bare nanoparticles. The same strategy was also used to improve the oral absorption of felodipine [48]. Kim et al. [49] carried out a detailed investigation of the oral availability of glycocholic acid-conjugated, solid fluorescent probe nanoparticles. The probe nanoparticle exhibited a significantly enhanced oral bioavailability with sustained absorption in a rat model. After oral administration, the following endocytosis into enterocytes at distal ileum via multivalent interaction with ASBT, the probe nanoparticles entered systemic circulation via the lymphatics (Figure 3). It was further demonstrated that a new oral route treatment regimen of docetaxel (DTX) using the ASBT targeting strategy was able to boost the anti-tumor efficacy of the drug [50]. The preparation consisted of DTX-loaded cationic solid lipid nanoparticles coated with an anionic polymer conjugated with glycocholic acid. The resulting nanoparticles were actively absorbed in the distal ileum mediated by ASBT. The plasma DTX profile showed the sustained presence of the drug up to 24 h after a single oral dose and did not impair the functions of the immune system. This in vivo study showed that the nanoparticles not only inhibited the growth of existing tumors but also decreased tumor formation when administered prior to cancer cell inoculation. The cytotoxic T cell population increased while the populations of tumor-associated macrophage and regulatory T cell decreased with this treatment regimen. The low-dose daily oral treatment may help patients with intermittent maintenance therapy and prevent tumor recurrence.

SMVT/SLC5A6 is an important transporter expressed in mammalian cells [51,52], which is responsible for the intestinal absorption of the water-soluble vitamins biotin, pantothenate, and lipoic acid. Biotinylated nanoparticles were developed to target this transporter as a means to increase the oral absorption of the drugs that have poor oral bioavailability. Oridonin is a natural compound with anti-inflammation and anti-cancer activities but displays low solubility and limited oral absorption. Zhou et al. [53] prepared biotinylated nanostructured lipid carriers to enhance the oral absorption of oridonin. Both the biotinylated and unmodified nanoparticles showed increased oral bioavailability of oridonin, but the oral absorption of biotinylated nanoparticles was significantly greater than that of the unmodified nanoparticles.

Vitamin C, also called ascorbic acid, is a water-soluble vitamin, and the transporters in SLC23 are responsible for its absorption in the intestine [54]. SVCT1 (SLC23A1) is primarily expressed in epithelial tissues such as the small intestine, where it mediates the absorption of vitamin C from dietary sources. Luo et al. [23] designed functional ascorbate-conjugated NPs (As-PLGA NPs) for enhanced oral paclitaxel delivery. As-PLGA NPs displayed increased uptake in Caco-2 cells, and the uptake process was Na^+^-dependent and was inhibited by free vitamin C, indicating that SVCT1 was involved in the internalization of As-PLGA NPs. The in vivo study confirmed that As-PLGA NPs enhanced the intestinal absorption compared to the bare nanoparticles. As such, SVCT1 holds potential as a target for oral delivery of therapeutic drug-loaded pharmaceutical nanocarriers.

### 2.2. Increasing the Oral Absorption of Macromolecules

Although the oral route is the safest and most practical to administer, it is generally considered not suitable for macromolecular drugs such as polysaccharides, oligopeptides, proteins, and nucleic acids. The luminal contents in the gastrointestinal tract get subjected to acidic pH in the stomach and a multitude of digestive enzymes (e.g., glycosidases, peptidases, proteases, nucleases, esterases, lipases, as well as detergents) in the intestine. Drugs of a macromolecular nature are likely degraded or denatured under these conditions, thus compromising their therapeutic efficacy. Additionally, the nutrient transporters expressed in the intestine do not normally accept macromolecules as their transportable substrates, thus reducing the absorbability of such macromolecular drugs. Furthermore, the intestinal barrier usually shows low permeation towards macromolecules, thereby resulting in a restricted oral absorption for these drugs via non-carrier-mediated processes. For these reasons, the delivery of macromolecular drugs via the oral route has proved to be extremely challenging. This necessitates the design of new approaches and the development of new delivery technologies for success in the oral administration of macromolecular drugs.

While ASBT/SLC10A2 has been utilized as a delivery target for small molecules, there is some evidence that this transporter also has potential in the delivery of large molecules if chemically modified appropriately. Lee et al. [55] conjugated deoxycholic acid to heparin for improved oral absorption via ASBT. As expected, the resultant bile acid-conjugated heparin exhibited improved bioavailability compared to free heparin. Interestingly, these investigators found that the solvent used to dissolve the drug had a significant impact on the absorption process. When dissolved in an organic solvent (e.g., dimethyl sulfoxide), the absorption of the bile acid-conjugated heparin increased almost 10-fold compared to that of free heparin, but the absorption was much slower when an aqueous solution was used to dissolve/disperse the drug [56,57]. This could simply be because ASBT recognizes only completely solubilized bile acid conjugates as substrates. Dispersing the highly hydrophobic bile acid conjugates in an aqueous solution prevents their recognition by and subsequent interaction with the transporter for eventual transport. Al-Hilal et al. synthesized several oligomers of deoxycholic acid and examined their affinity towards ASBT [58]. A tetrameric form of deoxycholic acid showed the highest affinity. Therefore, they conjugated this tetramer to heparin to enhance oral absorption [59]. In vivo studies showed that this conjugate could significantly increase the ASBT-mediated absorption in the intestine, evident from the markedly decreased coagulation-dependent tropism of fibrinogen [59,60]. Additional modification of tetramer-conjugated heparin with taurocholic acid, a different bile acid, improved the biological half-life in the circulation [61].

The same transporter has also been used for the oral delivery of insulin. Lee et al. were the first to develop an ASBT-targeted insulin delivery system for oral application [62]. They modified the recombinant human insulin by covalently attaching deoxycholic acid, and showed the suitability of such modified insulin for increased ASBT-mediated oral absorption. In a recent study, Fan et al. loaded insulin into deoxycholic acid-conjugated chitosan-constructed nanoparticles (DNPs) (Figure 4) [63]. In vitro studies showed that these bile acid-conjugated DNPs were internalized by enterocytes via ASBT-mediated endocytosis. DNPs bypass the endo-lysosomal pathway and thus protect the encapsulated insulin from biological degradation. Furthermore, DNPs also interact with the cytosolic bile acid-binding protein within the enterocytes, which facilitates the intracellular trafficking from the apical compartment to the basal compartment, with subsequent release of loaded insulin across the basolateral membrane. In pharmacokinetic studies, it was found that DNPs administered in the form of enteric-coated capsules had an oral bioavailability of 16% in an animal model of type I diabetes [63]. Zhang et al. developed cholic acid-conjugated nanoparticles (CCNPs) using a chitosan derivative and hydroxypropyl methylcellulose phthalate [64]. These CCNPs were designed to protect insulin from denaturation and degradation in the gastrointestinal tract, facilitate the intestinal absorption of the encapsulated insulin via ASBT-mediated uptake, and prolong the half-life of insulin through the cholic acid-guided enterohepatic circulation. As expected, the nanoparticles significantly increased the oral absorption of insulin, and successfully corrected the hyperglycemia for more than 24 h in diabetic mice. The pharmacological bioavailability of CCNPs was about 30% compared with that of insulin injected subcutaneously. These studies provide evidence for the successful application of ASBT-targeted NDDS for oral insulin delivery. Even though the therapeutic action of insulin was demonstrated with these bile acid-modified nano-delivery systems, there is no definitive information available on the route of entry of these nanosystems into the systemic circulation. Fan et al. of [63] surmised that theses nanoparticles entered into blood capillaries as one would expect free bile acids based on their enterohepatic circulation (Figure 4). In contrast, Kim et al. postulated that the ASBT-targeted nanoparticles are absorbed first into the lymphatic duct and then enter the systemic circulation as occurs for chylomicrons in the case of dietary lipids (Figure 3) [49]. Additional studies are needed to address this particular issue.

MCT1/SLC16A1 is ubiquitously expressed; it transports many types of monocarboxylates, such as pyruvate, lactate, α-keto acids arising from branched-chain amino acids, and the ketone bodies (acetoacetate, β-hydroxybutyrate) [65]. MCT1 is expressed on the apical side of enterocytes and is responsible for the absorption of dietary monocarboxylates (e.g., lactate) and bacterial fermentation products (e.g., acetate, propionate, and butyrate). Studies have shown that MCT1 expression is altered in the intestine and colon under inflammatory conditions [66]. Wu et al. conjugated butyrate on the surface of classical PEG NPs (Bu-PEG NPs) to enhance the oral absorption of insulin [67]. Butyrate-modified NPs could interact with MCT1, resulting in enhanced cellular uptake. In vivo studies showed that the insulin-loaded Bu-PEG NPs exhibited 2.9-fold higher oral bioavailability and a much stronger hypoglycemic response in diabetic rats compared to bare nanoparticles. These results suggested the potential of butyrate functionalization in NDDS design to improve the intestinal insulin absorption. A phenomenon of “easy entry, hard transcytosis” is normally observed in targeted oral drug delivery, indicating limited migration from apical to the basal side following a fairly easy entry from the lumen into the cells across the apical membrane. Further studies of the transport mechanisms of Bu-PEG NPs were carried out with the intestinal cell line Caco-2 (Figure 5) [68]. The results of these studies showed that after binding to MCT1, the nanoparticles went through the endo-lysosomal pathway, endoplasmic reticulum/Golgi recycling routes, and microtubule-dependent shuttling within the cell, and then exocytosed across the basolateral membrane. They found that increased hydrophobicity could facilitate nanoparticle transcytosis. Interestingly, when the basal expression of MCT1 was increased by leptin, the basal exocytosis and transcytosis of nanoparticles significantly increased. It is important to mention that the functionalization of the nanoparticles with butyrate involved conjugation via the carboxyl group of butyrate, thus eliminating the anionic carboxyl group in butyrate. Therefore, it is surprising that MCT1 played a role in the absorption of such nanoparticles. Further work is needed to address this issue in a more detailed manner. β-Hydroxybutyrate is also a transportable substrate for MCT1, but the functionalization of nanoparticles with this substrate can be achieved via the hydroxyl group without masking the anionic carboxylate group, thus making it a straightforward substrate for MCT1. This approach has been used successfully to target nanoparticles to MCT1 [69].

In addition, valine-conjugated PLA-PEG nanoparticles were prepared for oral oxytocin peptide delivery [31]. The addition of valine to PLA-PEG enables the peptide transporter PepT1 in the intestinal apical membrane to recognize the modified nanoparticle. This is similar to the effects of modifying the nucleosides acyclovir or ganciclovir with valine to promote their absorption via PepT1. Pharmacokinetic studies showed that the AUC of oxytocin peptide-loaded valine-conjugated nanoparticles increased 6-fold compared to that of the free oxytocin peptide, indicating that PepT1-targeting is a promising approach for oral peptide delivery in the form of nanoparticles. Similarly, the multivitamin transporter SMVT/SLC5A6 in the intestinal absorptive cells was targeted successfully for drug delivery by attaching biotin to liposomes, which resulted in increased oral absorption of insulin encapsulated within the liposomes [70]. As expected, biotinylated liposomes displayed a significant hypoglycemic effect after oral administration, and the bioavailability was up to 5-fold greater than that of the unmodified liposomes.

## 3. Transporter-Targeted NDDS for Enhancing BBB Permeation

Delivery of drugs to the brain has always been a challenge because of the blood–brain barrier (BBB) and blood–cerebrospinal fluid barrier (BCSFB). However, it has become obvious that the endothelial cells that comprise the BBB express numerous transporters to deliver essential nutrients to the brain cells and that such transporters could be exploited for drug delivery. Approximately 300 SLC transporters have been identified in the endothelial cells of BBB and epithelial cells of BCSFB [71]. Recently, transporter-targeted NDDS have been designed to enhance the permeability of loaded drugs across BBB. In this section, we will introduce the advances in the area of transporter-targeted NDDS, not only for enhanced BBB permeation but also for glioma therapy.

GLUTs are responsible for the transport of glucose from the circulation into target cells. These are facilitative transporters and the direction of transport is dictated simply by the direction of the glucose concentration gradient. GLUT1/SLC2A1 is a representative member of the GLUT family and is expressed ubiquitously. Specifically, it is responsible for glucose transfer across BBB. Therefore, glucose-conjugated NDDS were designed to study the potential of GLUT1 as a target for brain drug delivery. Anraku et al. prepared a self-assembled supramolecular nanocarrier modified with glucose on the surface [72]. The surface-attached glucose could interact with GLUT1 in brain capillary endothelial cells, thus facilitating the nanocarrier to cross BBB and enhancing the delivery of drugs in such nanocarriers into the brain. Manipulating the glucose density on the surface of the nanocarrier could also control its distribution within the brain. Xie et al. investigated the effect of linker length between glucose and nanoparticles on the performance of glucose-conjugated nanoparticles [41]. They prepared a series of glucose-modified liposomes using PEG with different chain lengths as the linkers. The qualitative and quantitative biodistribution assay in mice showed that the targeted liposomes using PEG1000 as the linker achieved the highest brain accumulation.

Tumor cells exhibit an abnormally increased demand for glucose due to metabolic reprogramming that favors aerobic glycolysis; glioma cells are no exception to this general rule. The increased demand for glucose in these cells is achieved by the upregulation of GLUT1 in the plasma membrane, thus facilitating increased delivery of glucose into the cells. The highly expressed GLUT1 in both BBB and glioma cells provides an ideal target for drug delivery for selective treatment of glioma. A proof-of-concept study was carried out by Jiang et al. [73] using 2-deoxy-D-glucose-functionalized PEGylated nanoparticles (DGlu-NPs). Compared to non-modified NPs, DGlu-NPs significantly increased the uptake of the NPs in BBB endothelial cells; this uptake of functionalized NPs was inhibited by free glucose, providing evidence for the involvement of GLUT1 in the uptake process of DGlu-NPs. In vivo studies showed that DGlu-NPs could selectively accumulate at the sites of glioma and exhibit efficient anti-glioma efficiency. Besides 2-deoxy-D-glucose, D-glucosamine [74] and dehydroascorbic acid [75] were also used as ligands of GLUT1-targeted NDDS for BBB and glioma dual-targeting capabilities. The GLUT1mediated dehydroascorbic acid transport is unidirectional, which is different from that of most other substrates of GLUT1. This unique feature of dehydroascorbic acid transport is explained based on the fact that dehydroascorbic acid, once inside the cell following its transport via GLUT1, is reduced immediately into ascorbic acid and gets trapped within the cells. Ascorbic acid is not a substrate for GLUT1; therefore, the intracellular reduction of dehydroascorbic acid is functionally coupled to cellular entry, and this process maintains the directionality of GLUT1-mediated transfer of dehydroascorbic acid. This feature makes dehydroascorbic acid-functionalized nanoparticles more efficient for glioma-targeted delivery. Shao et al. prepared the dehydroascorbic acid-modified nanoparticles with an additional disulfide bond crosslinking strategy, and successfully stabilized the nanostructure in systemic circulation and achieved a redox-responsive drug release in the tumor microenvironment [75]. Glutathione was used to reduce dehydroascorbic acid intracellularly, thus promoting the redox-responsive uptake of the GLUT1-targeted nanoparticles into cells.

Choline is an important nutrient involved in the synthesis of the membrane phospholipid phosphatidylcholine and the neurotransmitter acetylcholine. Choline transporter (ChT/SLC5A7) expressed in BBB is responsible for the brain absorption of choline. However, both the positively charged quaternary ammonium group, as well as the hydroxyl group in the structure of choline, are necessary for recognition by ChT as its substrate [76]; this restricted the potential of choline for modification of the delivery carriers because of the limitations in the conjugation process without involving the hydroxyl group and the quaternary ammonium group. Li et al. designed a series of derivatives of choline and tested their affinity towards ChT [77]. They conjugated the derivative that had the highest affinity onto dendrimers (DGL-PEG-CD) for brain-targeted plasmid delivery. In vivo studies showed that DGL-PEG-CD displayed significantly enhanced brain accumulation compared to that of conventional dendrimers [77]. Gliomas also express high levels of ChT to increase choline delivery into cancer cells for endogenous synthesis of phospholipids needed for membrane biogenesis [78]. In the follow-up study, the same investigators used DTPA-Gd, a contrast agent, to modify the ChT dual-targeting NPs for both drug delivery and precise diagnosis of glioma and monitoring of the treatment outcome [79]. The nanoprobe could cross the BBB and specifically accumulate in glioma cells via ChT. Under magnetic resonance imaging (MRI), the margins of the tumor were clearly identified by the accumulation of the nanoprobe, distinguishing it from the surrounding normal tissue. This nanoprobe provided a precise delineation of tumor margins for efficient surgical resection. The same choline derivative was also used to fabricate a co-delivery system of a plasmid encoding TRAIL and doxorubicin for glioma therapy [80]. The enhanced antitumor activities and precise tumor detection were confirmed in vivo using a mouse xenograft model. The choline derivative-modified micelles could also effectively suppress glioma growth and enhanced treatment outcome [25].

LAT1/SLC7A5 is an amino acid transporter that mainly transports branched-chain amino acids and aromatic amino acids. Its expression is increased in many cancers for the increased amino acid influx, and it represents the primary transporter in BBB for the delivery of amino acids to the brain [81]. Therefore, LAT1 could be utilized as an ideal target for glioma drug delivery by dual-targeting both at the level of BBB and at the level of glioma. Kharya et al. examined the feasibility of this strategy with phenylalanine-coupled solid lipid nanoparticles (PSLNs) [82]. The PSLNs showed increased drug distribution inside glioma cells. However, it should be mentioned that the free amino group and carboxyl group on the α-carbon atom of the amino acid are critical for the recognition of the substrate by LAT1 [83]. Nonetheless, in the design reported by Kharya et al. [82], the α-carboxyl group of phenylalanine was used to link nanoparticles, thus potentially interfering with the interaction of the phenylalanine-conjugated nanoparticles with LAT1. Even though the strategy was designed with a focus on LAT1 as the target, the involvement of the transporter in the uptake of PSLNs was not investigated with appropriately designed experiments. To avoid this pitfall, Li et al. conjugated the γ-carboxyl group of glutamate to the surface of liposomes to develop LAT1-targeted liposomes [84]. With this experimental approach, two important goals were achieved. First, as glutamate is not a substrate for LAT1 because of the extra anionic carboxyl group on the γ-carbon atom, conjugation of glutamate to liposome using the γ-carboxyl group masks this anionic group. Second, the amino group and the carboxyl group in the α-carbon atom are kept unmodified. Thus, the resultant glutamate-conjugated liposomes are recognized by LAT1. The glutamate-conjugated liposomes significantly increased the cellular uptake and cytotoxicity of docetaxel in C6 glioma cells compared to unmodified liposomes. In vivo studies showed that glutamate-conjugated liposomes could effectively cross the BBB and selectively distribute into the brain. Similar results were obtained with L-DOPA-conjugated liposomes, again targeting LAT1 in BBB and glioma [85].

MCT1 is expressed on the luminal membrane of brain capillary endothelial cells where it plays an essential role in transferring lactate and ketone bodies to the brain [71]. Venishetty et al. explored β-hydroxybutyrate-grafted docetaxel-loaded solid lipid nanoparticles to enhance the drug distribution to the brain [69]. The uptake of β-hydroxybutyrate-grafted nanoparticles in brain endothelial cells was significantly increased in comparison with unmodified nanoparticles. The uptake of β-hydroxybutyrate-conjugated nanoparticles could be suppressed by free β-hydroxybutyrate, suggesting an MCT1-mediated uptake mechanism.

SVCT2/SLC23A2 is also a transporter for vitamin C, but its tissue expression pattern is relatively broad [54]. It is highly expressed in brain microvascular endothelial cells to mediate the transportation of vitamin C across the BBB. Salmaso et al. investigated the potential of SVCT2 as a target for brain drug delivery with vitamin C-conjugated micelles [86]. Significantly increased uptake of vitamin C-conjugated micelles was observed in vitro, and the uptake was effectively abolished in the presence of free vitamin C, implicating SVCT2 in the uptake process.

Brain cells do not utilize β-oxidation of fatty acids to any significant extent to generate energy, but acetyl-L-carnitine plays a critical role in brain function and energy status [87,88]. Thus, carnitine is essential for brain metabolism even without its need for fatty acid oxidation. OCTN2 is the transporter for carnitine; it is highly expressed in brain capillary endothelial cells in the BBB and facilitates the transfer of acetyl-L-carnitine from the systemic circulation to the brain across the BBB. In addition, OCTN2 is also expressed in glioblastoma multiforme T98G cells. As such, studies in our laboratory explored the potential of L-carnitine-conjugated nanoparticles to facilitate permeation across the BBB and also simultaneously targeting glioma cells [89]. The effect of PEG spacer length on the OCTN2 targeting efficacy was also investigated. L-Carnitine conjugation could significantly increase the uptake of nanoparticles in BBB endothelial cells and glioma T98G cells. Compared to free paclitaxel, paclitaxel-loaded L-carnitine-conjugated nanoparticles showed a significantly increased anti-glioma activity. Specifically, when the PEG space was 1000, L-carnitine-conjugated nanoparticles displayed the best targeting efficacy, which was consistent with other reports [41]. In vivo studies demonstrated that L-carnitine-conjugated nanoparticles could effectively accumulate in the brain, confirming the potential of OCTN2 for brain targeting and glioma drug delivery.

## 4. Transporter-Targeted NDDS for Drug Delivery into Tumor Cells

SLC transporters play a critical role in the uptake of nutrients in mammalian cells. This phenomenon is particularly relevant to cancer cells, which exhibit an abnormally increased demand for nutrients to support their malignant growth. This necessity in cancer cells is satisfied with the upregulation of various nutrient transporters. Examples of the transporters that are expressed at high levels in cancer cells include GLUT1, LAT1, ASCT2, ATB^0,+^, and SLC7A11, among others. These transporters could be utilized as excellent targets not only for starvation therapy by blocking the function of these transporters but also for tumor-selective delivery of drugs/diagnostics by selectively targeting these transporters [26,27,90].

As one of the most significant hallmarks of cancer cells, aerobic glycolysis (also called the Warburg effect) is well-recognized by the high demand for glucose. GLUT1 is expressed at a much higher level in cancer cells than in normal cells to support the increased demand for glucose. This forms the basis for the use of ^18^F-fluoro-2-deoxy-D-glucose (^18^FDG) in positron emission tomography for in vivo tumor diagnosis. This glucose derivative is a positron emitter and also a substrate for GLUT1. As tumor cells express high levels of GLUT1, ^18^FDG accumulates in tumor cells more than in the surrounding normal cells. Once inside the cells, ^18^FDG gets phosphorylated by hexokinase and the resultant phosphate derivative cannot go through glycolysis any further because of the 2-deoxy group and thus ends up trapped inside the cells. As a result, the positron signal is more pronounced at the site of the tumor compared to normal regions, thus helping in the precise detection of the tumor site in vivo. There are other applications of GLUT1 as well in the field of cancer biology. Shan et al. fabricated γ-Fe_2_O_3_ nanoparticles coated with dimercaptosuccinic acid and modified with 2-deoxy-D-glucose to target GLUT1 for precise diagnosis of the tumor by MRI [91]. The MRI T2 signal intensity in breast cancer significantly increased when treated with such nanoparticles, substantiating that the GLUT1-targeted nanoparticles could be applied for precise tumor imaging. Xiong et al. demonstrated the feasibility of this strategy in cervical cancer [92]. In another study, an MRI contrast agent targeting GLUT1 for tumor detection was developed; this was constructed based on paramagnetic gadolinium oxide (Gd_2_O_3_), coated with polycyclodextrin (PCD), and further modified with glucose (Gd_2_O_3_@PCD-Glu) [93]. The results showed that the acquired MRI T1 signal intensity was significantly increased in MDA-MB-231 cells treated with the Gd_2_O_3_@PCD-Glu NPs, indicating Gd_2_O_3_@PCD-Glu NPs could be applied as an MRI-targeted tumor agent to enhance GLUT1-overexpressed tumor theranostics. GLUT1-targeted NDDS strategy could also be used for tumor-targeting drug delivery. Zaritski et al. designed hydrolyzed galactomannan (hGM)-based amphiphilic nanoparticles for the selective tumoral accumulation of drugs in pediatric patient-derived sarcomas [94]. The nanoparticles could be internalized into the cells about 100% at 37 °C. In vivo studies confirmed that the intra-tumoral accumulation of nanoparticles correlated well with the expression level of GLUT1 in each patient-derived tumor. In another study, poly((D,L)lactic-glycolic)acid-star glucose (PLGA-Glc) polymer-based nanoparticles were prepared for tumor-targeted delivery of docetaxel [95]. PLGA-Glc NPs displayed significantly enhanced tumor-targeting abilities, which was attributed to the specific interaction with GLUTs. Li et al. designed a multi-functional nanoparticle by fabricating glucosyl ligands and bortezomib (BTZ) to the catechol groups in polydopamine (PDA) NPs in such a manner that the nanodevices will respond to the acidic pH in the tumor microenvironment (Figure 6) [96]. This multi-functional nanoparticle had a GLUT1-targeted, tumor microenvironment-responsive and near-infrared irradiation (NIR)-induced cytosolic drug delivery. These studies showed that the multi-functional nanoparticles could selectively accumulate at the tumor site facilitated by the acidic tumor microenvironment, and localize in subcellular endo-lysosomes of tumor cells, and also promote the release of BTZ in response to NIR irradiation. It was notable that a single dose application was sufficient to suppress the growth of 4T1 cell tumors in mice. Glucose-conjugated PAMAM was also developed for enhanced tumor targeting [96,97,98]. Another therapeutic application of GLUT1-targeting is in nucleic acid delivery. Guo et al. constructed a dehydroascorbic acid-conjugated and glutathione-responsive drug delivery system for effective anti-miR21 delivery and cancer therapy [24]. Yi et al. decorated a 20-nm gold nanoparticle with a glucose-installed poly(ethylene glycol)-block-poly(L-lysine) modified with lipoic acid to form a polo-like kinase 1 (PLK1) siRNA (siPLK1) nanocarrier [99]. Both methods displayed significantly enhanced anticancer efficacy attributed to GLUT1-mediated targeting. Interestingly, Cheng et al. used glucose-conjugated fluorescent gold nanoclusters as a probe for the quantitative analysis of cancer aggressiveness by determining the glucose metabolism rate of glucose-AuNCs in cancer cells [100]. This work provided a practical application of GLUT1-targeted gold nanoclusters. Abolhasani et al. used glucose-conjugated nanoparticles to block GLUT1 for starvation therapy [101]. They found that the survival rate of cancer cells was decreased ~45% after treatment of glucose-conjugated nanoparticles. Here the purpose of the glucose-conjugated nanoparticles was not for drug delivery but for blocking the function of GLUT1 and preventing the delivery of glucose to cancer cells. A follow-up study showed that the mRNA expression of GLUT1 was significantly upregulated after the first 24 h of treatment of glucose-conjugated nanoparticles, but decreased to 45% of control after 72 h, which coincided with the cell death. Collectively, the GLUT1 targeting strategy could be used to design functional NDDS for multiple benefits, including enhanced drug delivery, combination therapy, diagnostic imaging, and nutrient starvation in cancer treatment.

Glutaminolysis is another feature of metabolism unique to cancer cells [102]. Cancer cells display an increased demand for glutamine for growth and proliferation. The tricarboxylic acid cycle (TCA cycle) in cancer cells is primarily fueled by α-ketoglutarate derived from glutamine via glutaminolysis; glucose does not contribute much to the TCA cycle because most of the pyruvate resulting from glycolysis gets converted to lactate, a hallmark of the Warburg phenomenon. The increased demand for glutamine is associated with increased expression of glutamine transporters in cancer cells. One of the glutamine transporters that has caught attention is ASCT2/SLC1A5, which is an amino acid exchanger that mediates the Na^+^-coupled influx of glutamine into cancer cells in exchange for Na^+^-coupled efflux of some other neutral amino acid [27,102,103]. This transporter could be targeted for drug delivery into cancer cells. Wang et al. developed glutamine macromolecular analog polyglutamine (PGS) for the siRNA delivery in cancer therapy [104]. The molecular docking analysis confirmed SLC1A5 has a high affinity towards PGS. PGS/siRNA complexes accumulated remarkably in cancer cells, especially when the cells were deprived of glutamine. When the expression of SLC1A5 was decreased, the cellular uptake of PGS/siRNA complexes also decreased, indicating the critical role of SLC1A5 in PGS uptake. In a lung orthotopic tumor model, a hybrid siRNA (anti-Survivin and anti-MDR1 (siSM))-loaded PGS delivery system decreased the tumor growth, while concurrent administration of PGS/siSM and cisplatin synergistically enhanced this effect and significantly improved life span.

Three other amino acid transporters are highly expressed in tumor cells, including LAT1/SLC7A5, ATB^0,+^/SLC6A14, and xCT/SLC7A11 [102,105]. Li et al. conjugated glutamate on the PLGA nanoparticles to target the overexpressed LAT1 in tumor cells [106]. Glutamate-conjugated nanoparticles (conjugation done at the γ-carboxyl group) showed a significant increase in cellular uptake and cytotoxicity in HeLa and MCF-7 cells compared to the unmodified nanoparticles. Interestingly, they found that the internalized LAT1 coupled with nanoparticles could recycle back to the cell membrane, allowing continuous targeted drug delivery. Glutamate-conjugated nanoparticles exhibited enhanced tumor accumulation and antitumor effects also in a xenograft model. Ong et al. found that conjugation of L-DOPA to anisotropic gold nanoparticles could induce selective photothermal ablation of breast cancer cells and sensitize the cells to chemotherapy [107]. These studies confirmed the potential of LAT1 as a target for selective drug delivery to cancer cells.

Our laboratories have been focusing primarily on ATB^0,+^/SLC6A14. This transporter exhibits unique functional features ideal to satisfy the increased demands for amino acids in tumor cells [105,108]. It can transport 18 of 20 amino acids, the only exception being the anionic amino acids glutamate and aspartate; it is also highly concentrative due to the coupling of its transport function to transmembrane gradients of Na^+^ and Cl^−^, and membrane potential. Compared to normal tissues, ATB^0,+^ is highly expressed in many kinds of tumors, particularly in estrogen receptor-positive breast cancer [109], colon cancer [110], cervical cancer [111], and pancreatic cancer [105,108]. We have developed amino acid-conjugated liposomes for ATB^0,+^ targeting to enhance the anticancer efficacy [112]. Three different amino acids, glycine, aspartate, and lysine, were linked to liposomes and tested for their ATB^0,+^ targeting efficiency. Lysine-conjugated liposomes showed the best efficacy in uptake and cytotoxicity assay in HepG2 cells which have a robust expression of ATB^0,+^, but not in L929 cells which have low levels of ATB^0,+^. Lysine-conjugated liposomes also displayed selective accumulation in tumors and distinct anticancer effects in a xenograft model. The molecular dynamic simulation showed that the binding energy of lysine-conjugated liposomes to the transporter decreased in the presence of Na^+^ and Cl^−^, demonstrating that Na^+^ and Cl^−^ are necessary for the interaction of lysine-conjugated liposomes with the transporter. Free aspartate is not a substrate of ATB^0,+^, but it can be recognized by ATB^0,+^ when the β-carboxyl group was masked by the conjugation. Therefore, we also tested the potential of aspartate as a ligand for ATB^0,+^ targeting, which showed therapeutic improvement [113]. In a further study, we investigated in detail the uptake mechanism of lysine-conjugated liposomes [114]. ATB^0,+^-positive (MCF7 cells) and ATB^0,+^-negative (MB231 cells) cells were used. Lysine-conjugated liposomes displayed significantly increased uptake and cytotoxicity in MCF7 cells, but not in MB231 cells. The uptake of lysine-conjugated liposomes consisted of two steps: binding and internalization. The binding process of lysine-conjugated liposomes in MCF7 cells was dependent on the presence of Na^+^ and Cl^−^, and inhibited by glycine, lysine and α-methyl-D,L-tryptophan (a specific blocker of ATB^0,+^), indicating the involvement of ATB^0,+^. The internalization process was independent of lysine. Endocytosis was observed in the uptake of lysine-conjugated liposomes, and ATB^0,+^ protein was subjected to transient endosomal degradation with subsequent recovery. Moreover, lysine-conjugated liposomes also enhanced the uptake and cytotoxicity of gemcitabine in pancreatic cancer cells in an ATB^0,+^-dependent manner. These results confirmed that this approach represented a novel strategy for enhanced therapy of ATB^0,+^-positive cancers.

The transfer of L-carnitine across the plasma membrane in mammalian cells is mainly dependent on OCTN2/SLC22A5, a high-affinity transporter for L-carnitine [115]. ATB^0,+^/SLC6A14 could also transport L-carnitine, but with a lower affinity compared to OCTN2/SLC22A5 (Km ≈ 800 µM) [116,117]. It was found that both OCTN2 and ATB^0,+^ are expressed at a much higher level in human colon cancer cells than in normal colon cells [118]. Therefore, we tested the potential of L-carnitine-conjugated nanoparticles dually targeting OCTN2 and ATB^0,+^ for colon cancer therapy [118]. The results showed that L-carnitine-conjugated nanoparticles significantly increased the uptake and potentiated the anticancer efficacy of 5-fluorouracil cargo in colon cancer cells, but displayed decreased uptake and significantly less cytotoxicity in normal colon cells. The colocalization assay indicated that both OCTN2 and ATB^0,+^ were involved in the uptake of L-carnitine-conjugated nanoparticles. These findings suggest that nanoparticles targeted to OCTN2 and ATB^0,+^ have a great potential to deliver chemotherapeutic drugs for colon cancer therapy.

SMVT/SLC5A6 is also expressed at a higher level in tumors than in normal tissues. As such, this transporter has also been explored as a target for selective drug delivery to cancer cells. There have been several investigations examining the potential of biotinylated nano-vehicles for cancer therapy, including PAMAM [119,120], liposomes [121], nanoparticles [122], and micelles [123]. However, a majority of these studies only focused on the increased uptake in cancer cells and the enhanced accumulation at the tumor site. Few studies focused on the mechanisms involved in the uptake process per se. Aleandri et al. prepared biotinylated cubosomes to simultaneously transport anticancer drugs and fluorescent dye into cancer cells [124]. The increased uptake and cytotoxicity in HeLa cells were compromised when free biotin was introduced, indicating the participation of SMVT in the delivery of biotinylated cubosomes into these cells. A tumor-targeted gene delivery system has also been developed based on the SMVT-targeted NDDS [125]. These studies confirmed the effectiveness of SMVT as a target for enhanced tumor drug delivery.

## 5. Transporter-Targeted NDDS for Topical Ocular Drug Delivery

PepT1 is rich in the corneal and conjunctival epithelial cells, and therefore could be used to enhance the drug permeability by transporter-mediated translocation for enhanced ocular drug delivery [126]. Xu et al. constructed chitosan oligosaccharide-valylvaline-stearic acid (CSO-VV-SA) nanomicelles for topical ocular dexamethasone delivery [127]. Appropriate competition studies confirmed the involvement of PepT1 in the uptake of CSO-VV-SA nanomicelles, and there was also evidence that the topically applied nanomicelles entered the posterior segment mainly through conjunctival route. In vivo precorneal retention studies demonstrated that dexamethasone from the nanomicelles could be detected for more than 3 h in rabbit tears. These findings indicate that CSO-VV-SA nanomicelles have promise for ocular drug delivery.

OCTN1 and OCTN2 are also expressed in ocular epithelia and mediate the transport of L-carnitine in corneal and conjunctival epithelial cells [128,129]. Therefore, they could be used as targets for topical ocular drug delivery. Bongiovì et al. developed a nanomicelle conjugated with L-carnitine to deliver imatinib for the treatment of neovascular ocular diseases [130]. The imatinib-loaded L-carnitine-conjugated micelles displayed optimal particle size and mucoadhesive properties and were able to interact with the corneal barrier and promote imatinib trans-corneal permeation, thus exerting an inhibitory effect on a choroidal neovascularization process. However, the authors only investigated the potential of nanomicelles for topical ocular drug delivery but did not identify the actual target site. In another study, L-carnitine was found to protect human retinal pigment epithelial cells from oxidative damage [131]. These studies highlight the potential of L-carnitine-conjugated DDS for the treatment of ocular diseases.

## 6. Transporter-Targeted NDDS for Other Indications

PepT1 is highly expressed in the small intestine, where it mediates the absorption of di- and tripeptides as well as peptidomimetic drugs; it is not expressed in normal colonic epithelial cells. However, the transporter was found to be anomalously overexpressed in inflammatory colonic epithelial cells and macrophages [132,133], indicating that PepT1 could be a promising target for the delivery of drugs to treat colonic inflammation. Wu et al. modified a PLGA nanoparticle with lysine-proline-valine (KPV) tripeptide to load cyclosporine A (CyA), and the resultant PepT1-targeted NPs were further coated with Montmorillonite (MMT) and chitosan (CS) to reduce CyA leakage in the upper gastrointestinal tract and enhance nanoparticle adhesion to the inflamed colon [134]. In vivo studies then showed that the CyA-PLGA-KPV/MMT/CS nanoparticles (PKMCN) specifically accumulated in the inflamed tissues and were retained at these sites for up to 36 h; they also exhibited significant therapeutic efficacy, evidenced by the improvements in body weight, colon length, and disease activity index. Interestingly, the authors found that the blank nanoparticle without CyA also displayed marked therapeutic effects, suggesting the PepT1-mediated adhesion of PKMCN on the inflamed intestine benefits the healing process. In addition, this could also be due to the anti-inflammatory activity of the lysine-proline-valine tripeptide used to target PepT1.

GLUT4 is expressed only in cardiac muscle, skeletal muscle, and adipocytes, and could respond to insulin [135]. The translocation of GLUT4-containing vesicles is induced by the interaction of insulin with its cell-surface receptor and the resultant intracellular signaling pathways. Yeh et al. developed glucose-modified quantum dots (Glc-QDs) for theranostic application [136]. The uptake of Glc-QDs in insulin-stimulated C2C12 muscle cells was significantly increased compared to control cells without exposure to insulin, and this uptake could be inhibited by the free 2-deoxy-D-glucose, indicating the involvement of GLUT4.

The design of hepatotropic drug carriers is of great interest for the treatment of various liver disorders. The hepatitis B virus (HBV) shows pronounced efficacy to infect the human liver due to its strong affinity to hepatocytes. For decades, the specific target of HBV on the sinusoidal membrane of hepatocytes was unknown until the interaction of the virus with the human sodium-taurocholate cotransporting polypeptide (NTCP/SLC10A1) was identified [137]. Witzigmann et al. modified synthetic lipid-based nanoparticles with targeting peptides derived from the hepatitis B virus large envelope protein (HBVpreS) to specifically target NTCP/SLC10A1 on the sinusoidal membrane of hepatocytes [138]. NTCP-specific, ligand-dependent binding and internalization were confirmed in vitro. In vivo studies showed that these delivery systems could increase liver uptake, decrease accumulation in off-target tissues, and at the same time avoid clearance by the reticuloendothelial system by mimicking HBV targeting properties.

SMVT/SLC5A6 expressed in the liver has also been examined for liver targeting delivery. A biotinylated erythrocyte was developed by Mishra et al. to deliver methotrexate to the liver [139]. In this in vitro study, biotinylated erythrocytes displayed a significantly enhanced uptake than the unmodified counterparts. In vivo results showed that biotinylated erythrocytes accumulated in the liver and displayed a three-fold enhanced therapeutic index compared to the unmodified erythrocytes.

ASBT is responsible for the absorption of bile acids in the ileum for enterohepatic circulation, and bile acids play a critical role in aiding the digestion and absorption of dietary fat. Therefore, blocking ASBT could suppress bile acid reabsorption, and thereby decrease fat absorption, holding the potential to prevent obesity and hyperlipidemia. Park et al. developed a novel hydrophilic ASBT inhibitor by conjugating a tetrameric form of deoxycholic acid to polyacrylic acid to prevent the ileal absorption of bile acids in vivo and interrupt the enterohepatic circulation of bile acids [140]. The conjugate could effectively bind to ASBT in MDCK cells. In vivo, this inhibitor was able to inhibit high-fat diet-induced hyperlipidemia.

## 7. Conclusions

The field of membrane transporter-targeted NDDS has seen significant progress in recent years and the results have been highly encouraging in terms of its potential for various therapeutic purposes. Here we have reviewed the recent advances in transporter-targeted NDDS. A critical analysis of the current literature highlights the potential of transporter-targeted NDDS for enhanced and site-selective drug delivery. This includes areas related to oral drug delivery, brain/glioma-targeting, tumor-targeting, and ocular-targeting, among others. The transporters that have been investigated and evaluated for these purposes are listed in Table 1.

The plasma membrane transporters have their physiologic functions in facilitating the entry of selective substrates into cells. But in most cases, the transporters are capable of interacting with exogenous xenobiotics if there is sufficient structural resemblance to their physiologic substrates. In the past, the focus of this particular feature of the transporters has been on issues related to drug-drug interaction [21,141]. The possibility that the same apparent lack of strict substrate selectivity of many of these transporters could be exploited for beneficial purposes was not appreciated for a long time. Now we know that the broad selectivity of some of these transporters is good for the delivery of specific drugs into target cells. This can be achieved for certain drugs as themselves or in the form of prodrugs where the drugs are chemically modified to enhance the interaction with a given transporter. In the current review, we have highlighted how this feature can also be exploited for drug delivery via nanocarriers. The logic is simple. These nanocarriers can be chemically modified on their surface with appropriate ligands to specifically target a given transporter. Once this targeting is achieved successfully, the nanocarriers can be used as a vehicle to carry any therapeutic drug as cargo for delivery into target cells that express the given transporter on the cell surface. In this review, we have provided convincing evidence from published reports for the success of this approach for drug delivery.

The traditional strategy in nano-delivery systems to target transporters is to conjugate specific substrates to nanoparticles. But the same strategy could also be used to block the normal function of the transporters for therapeutic purposes. This is particularly apparent in cancer therapy because cancer cells rely on selective transporters to provide nutrients. If such transporters can be blocked with appropriately conjugated nanoparticles, cancer cells will undergo nutrient deprivation. This approach has the potential in subjecting cancer cells to growth arrest and/or cell death [101,104,140,142].

Transporter-targeted drug delivery has achieved considerable success in the pharmaceutical industry in the form of prodrugs. NDDS has been in development for several decades, but to date, only conventional nanoparticles have been approved by the US Food and Drug Administration (FDA) for therapeutic purposes. Transporter-targeted NDDS has shown excellent drug delivery and treatment efficacy in preclinical studies, but no formulation in this category has yet reached a stage sufficient for approval for use in the clinics. More work needs to be done in this area, but the rationale and the scientific premise remain unquestionable for the validity of this approach.

## Figures and Tables

**Figure 1 cancers-12-02837-f001:**
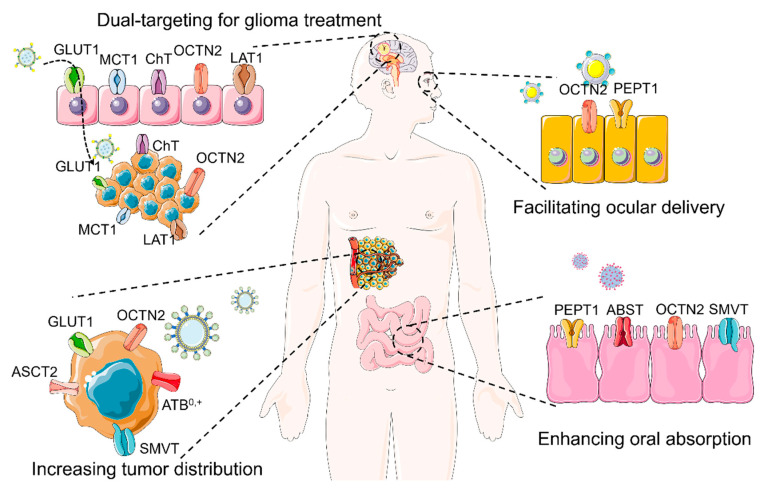
Transporter-targeted nano-drug delivery systems for enhanced and site-specific drug delivery. GLUT1, facilitative glucose transporter-1 (SLC2A1); MCT1, monocarboxylate transporter-1 (SLC16A1); ChT, choline transporter (SLC5A7); OCTN2, novel organic cation transporter-1 or organic cation/carnitine transporter (SLC22A5); LAT1, L-amino acid transporter-1 (SLC7A5); ASCT2, Alanine-Serine-Cysteine Transporter-1 (SLC1A5); SMVT, sodium-coupled multivitamin transporter (SLC5A6); ATB^0,+^, amino acid transporter B^0,+^ (SLC6A14); PEPT1, peptide transporter (SLC15A1); ABST, apical bile salt transporter (SLC10A2).

**Figure 2 cancers-12-02837-f002:**
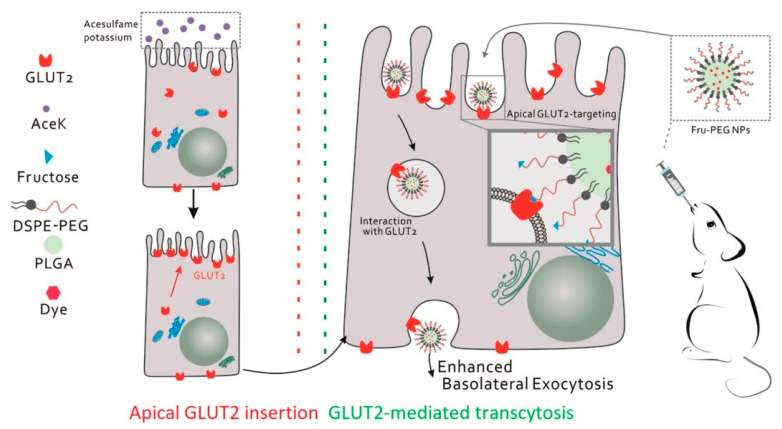
Acesulfame potassium (AceK) induces the apical recruitment and density of GLUT2 in intestinal epithelial cells, which activates the transcytosis of fructose-conjugated nanoparticles (Fru-PEG NPs) across the enterocyte. Reproduced with permission from [35]; Copyright 2020 Elsevier Ltd.

**Figure 3 cancers-12-02837-f003:**
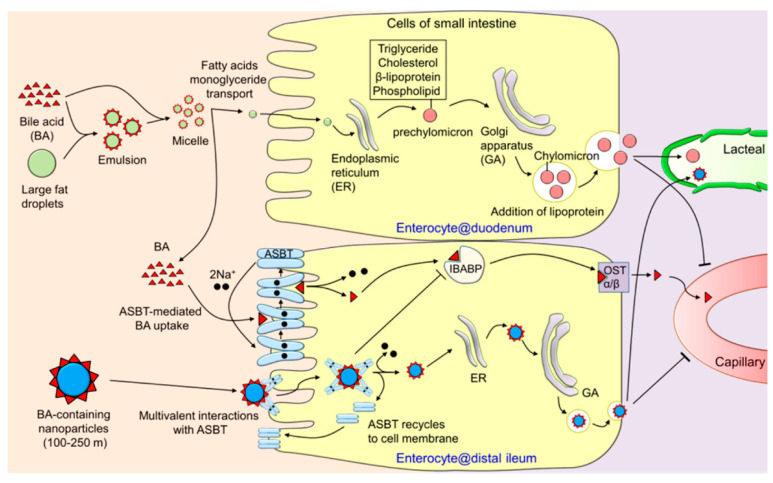
After oral administration, the bile acid-conjugated nanoparticles are endocytosed into enterocytes at the distal ileum via multivalent interaction with ASBT and subsequently appear in the systemic circulation via the lymphatic system. Reproduced with permission from [49]; Copyright 2018 American Chemical Society.

**Figure 4 cancers-12-02837-f004:**
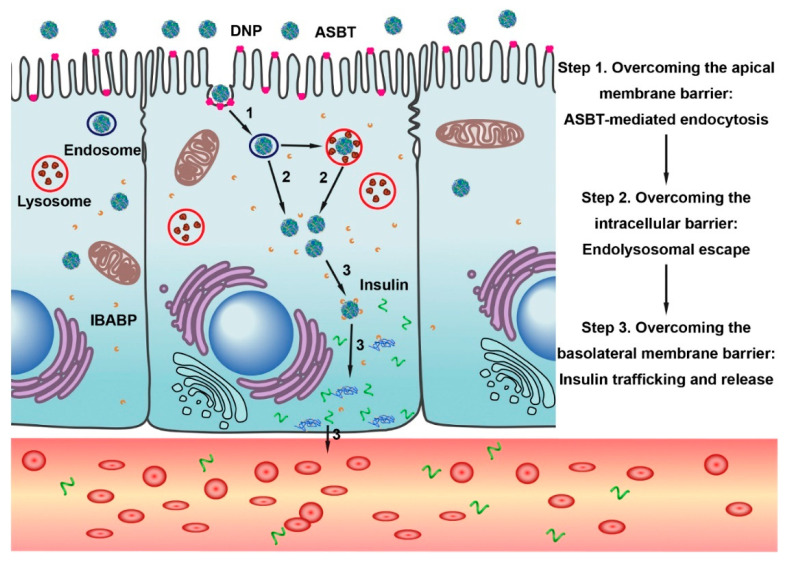
Schematic illustration of insulin-loaded DNPs overcoming multiple barriers of the intestinal epithelium by exploiting the bile acid pathway for enhanced oral absorption. Reproduced with permission from [63]; Copyright 2018 Elsevier Ltd.

**Figure 5 cancers-12-02837-f005:**
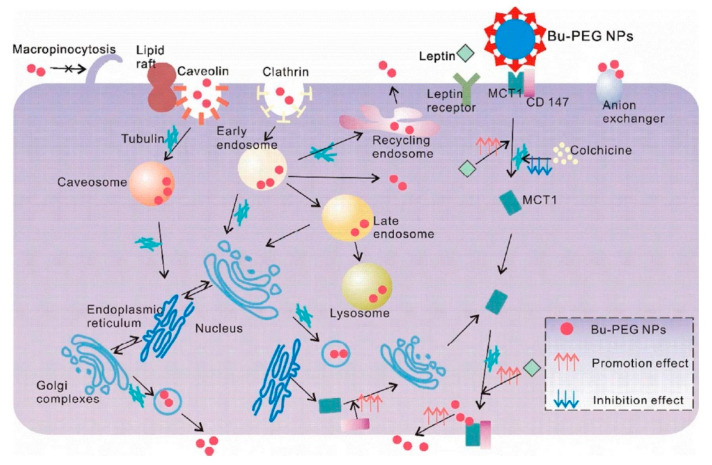
Transport pathways of butyrate-conjugated nanoparticles (Bu-PEG NPs) and factors that influence each stage of transcytosis. Reproduced with permission from [68]; Copyright 2018 American Chemical Society.

**Figure 6 cancers-12-02837-f006:**
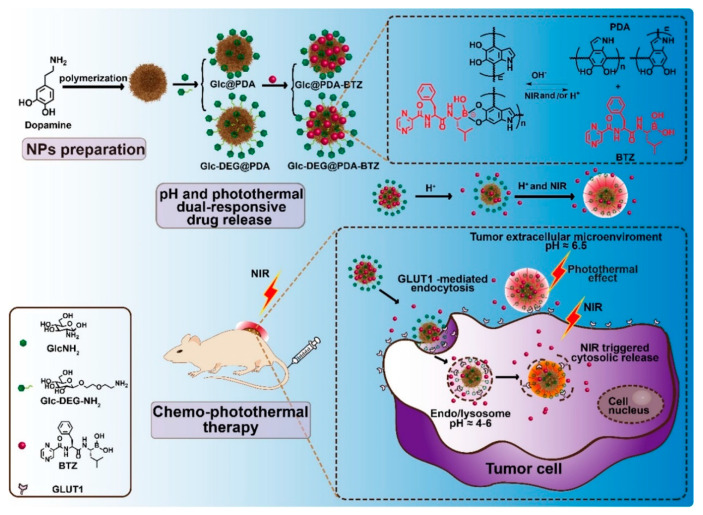
Schematic illustration of the construction of glucose-functionalized polydopamine (PDA) nanoparticles with pH and photothermal dual-responsive and photothermally triggered cytosolic drug delivery properties for the GLUT1-targeting chemo-phototherapy. Reproduced with permission from [96]; Copyright 2020 Elsevier Ltd.

**Table 1 cancers-12-02837-t001:** Plasma membrane transporters that have been evaluated for targeted drug delivery in the form of nanodevices.

Transporter	HUGO Nomenclature	Target for Drug Delivery
ASCT2	SLC1A5	Tumors
GLUT1	SLC2A1	Blood–brain barrier Glioma
GLUT2	SLC2A2	Oral absorption
GLUT4	SLC2A4	Muscle, heart, adipocyte
SMVT	SLC5A6	Oral absorption Tumors, Liver
ChT	SLC5A7	Blood–brain barrier Glioma
ATB^0,+^	SLC6A14	Tumors
LAT1	SLC7A5	Blood–brain barrier Tumors
ASBT	SLC10A2	Oral absorption
PepT1	SLC15A1	Oral absorption Ocular delivery Colitis
MCT1	SLC16A1	Oral absorption Blood–brain barrier
OCTN2	SLC22A5	Oral absorption Blood–brain barrier Glioma Ocular delivery
SVCT2	SLC23A2	Blood–brain barrier

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
