# Peer review of "Transporter-Targeted Nano-Sized Vehicles for Enhanced and Site-Specific Drug Delivery"

_cancers, 2020, doi:10.3390/cancers12102837_

Round 1

Reviewer 1 Report

The review summaries quite a good recent literatures on delivery of chemotherapeutic drugs to cancer cells by targeted nano-delivery systems and can be accepted after minor english correction.

Author Response

We went through the entire manuscript very carefully and corrected all the grammatical errors. Thank you for pointing out the deficiency in the manuscript in this area.

Reviewer 2 Report

Dear author,

I have read the manuscript with high interest. The review is very timely and describes major developments on transporter targeted NDDS for site specific delivery. Hence I am recommending to publish after minor revision.

  1. A comprehensive table with major examples from different categories will be an excellent addition to the paper.
  2. What are the major hurdles for this and how these can be addressed for better clinical translation. 
  3. Any toxicity issues which need to be highlighted. Can these approaches reduce the overall toxicity?
  4. There are minor typo and grammatical correction needed overall 

Author Response

We have now added a Table at the end of the manuscript, listing the transporters that have been evaluated thus far for the targeted delivery of drugs in the form of nano devices.

The hurdles and toxicity issues have already been included and discussed in the Introduction and Conclusions sections of the manuscript. We could not think of any additional ideas for inclusion. This is because the field of using plasma membrane transporters for targeted drug delivery in the form of nano devices is new. We believe that as the field advances further, we will come to know if there are any potential hurdles and toxicity issues in the approach. Only then, steps could be undertaken to see if such hurdles and issues could be resolved.

We have edited the manuscript carefully and corrected typographical and grammatical errors. 

Reviewer 3 Report

Recent years, NDDS have aroused attention due to their potential to minimize side-effects, reduce dosage and customized targeting to specific cell types or tissues.

Some NDDS formulated drugs have been approved to use in clinical while many others are undergoing clinical trials. However, no Transporter-targeted Nano-Sized Vehicles have moved to clinical trial. This area is really a point of interest and this review came just in time.

This paper is well written and covered the major aspects of this area very well.

Minor suggestion:

  1. Recently, I came to an interesting paper : Norepinephrine-Transporter-Targeted and DNA-Co-Targeted Theranostic Guanidines. See if there is any chance you could also discuss this one in the cancer.
  2. The other suggestion would be there are too many transporters, its better to make a tale to list them and give readers a clear overall idea.

Personally, I insist on suggestion 2 as necessity and put the suggestion 1 as just option.

Author Response

We checked the publication indicated by the reviewer. The study used the norepinephrine transporter as a drug delivery system. But the study has nothing to do with nano devices. It simply evaluated the transporter for the delivery of a modified substrate into cells. Therefore, we think that inclusion of this study is not warranted in this review that focuses on the use of transporters for drug delivery in the form of nano devices. We sincerely hope that the reviewer could concur with our view.

We have now added a Table at the end of the manuscript, listing all the transporters that have been evaluated thus far for their utility in targeted drug delivery in the form of nano devices.

Reviewer 4 Report

The paper by Kou and colleagues is well written and reviews all the literature on nano particle transporters within cells.
The English language is correctly expressed.
The figures are clear and the legends adequate.
The literature is up-to-date and the references correctly inserted in the text.

Author Response

This reviewer did not note any specific deficiency or concern.